# The Abundance and Taxonomic Diversity of Filterable Forms of Bacteria during Succession in the Soils of Antarctica (Bunger Hills)

**DOI:** 10.3390/microorganisms9081728

**Published:** 2021-08-13

**Authors:** Alina G. Kudinova, Andrey V. Dolgih, Nikita S. Mergelov, Ilya G. Shorkunov, Olga A. Maslova, Mayya A. Petrova

**Affiliations:** 1Institute of Molecular Genetics of National Research Centre «Kurchatov Institute», Akademika Kurchatova Square 2, 123182 Moscow, Russia; elvi.23@mail.ru; 2Institute of Geography, Russian Academy of Sciences, Staromonetnyy Lane 29, 119017 Moscow, Russia; dolgikh@igras.ru (A.V.D.); mergelov@igras.ru (N.S.M.); shorkunov@igras.ru (I.G.S.)

**Keywords:** ultramicrobacteria, dormant cells, next-generation sequencing, bacterial community structure, mesomorphology and micromorphology of soil

## Abstract

Previous studies have shown that a significant part of the bacterial communities of Antarctic soils is represented by cells passing through filters with pore sizes of 0.2 µm. These results raised new research questions about the composition and diversity of the filterable forms of bacteria (FFB) in Antarctic soils and their role in the adaptation of bacteria to the extreme living conditions. To answer such questions, we analyzed the succession of bacterial communities during incubation of Antarctic soil samples from the Bunger Hills at increased humidity and positive temperatures (5 °C and 20 °C). We determined the total number of viable cells by fluorescence microscopy in all samples and assessed the taxonomic diversity of bacteria by next-generation sequencing of the 16S rRNA gene region. Our results have shown that at those checkpoints where the total number of cells reached the maximum, the FFB fraction reached its minimum, and vice versa. We did not observe significant changes in taxonomic diversity in the soil bacterial communities during succession. During our study, we found that the soil bacterial communities as a whole and the FFB fraction consist of almost the same phylogenetic groups. We suppose rapid transition of the cells of the active part of the bacterial population to small dormant forms is one of the survival strategies in extreme conditions and contributes to the stable functioning of microbial communities in Antarctic soils.

## 1. Introduction

Bacteria have a wide variety of survival strategies under adverse conditions such as low temperatures, ultraviolet radiation, low humidity, and nutrient deficiencies [1,2]. There is a hypothesis that in the soil, especially in the conditions of the nitrogen and carbon limitation, as well as in habitats with extremely low temperature, a significant part of the cells is dormant, which is often characterized by a decrease in cell size [3,4]. In addition, the bacterial strains isolated from permafrost are more prone to go into a dormant state in the form of ultra-small cells than strains of the same genera isolated from soils of temperate latitudes [5].

It is known that among the cells obtained by filtration through filters with a pore size of 0.2 µm, there may be: (I) ultramicrocells of common bacteria that have a decreased size due to external factors (stress, lack of nutrients); (II) obligate ultramicrobacteria (UMB) that maintain small cell volumes (<0.1 µm^3^) regardless of their growth conditions; (III) facultative UMB that contain a small proportion of larger cells with a cell volume of >0.1 µm^3^ in the population; (IV) slender filamentous bacteria capable of “squeezing” through the filter pores [6]. Given the above, in this article, we use the term filterable forms of bacteria (FFB) to refer to the cell fraction obtained by filtering.

Ultramicrocells whose volume does not exceed 0.1 µm^3^ are found in various natural environments: aquatic, hyperthermophilic, inside living organisms, including humans, on the surface of rocks and in the soils [7]. The small dormant forms of clinically significant bacteria *Mycobacterium tuberculosis* have been studied most fully and it has been proved that they cause the latent form of infection [8]. However, there are not so many works devoted to the study of FFB in natural habitats and a major part of such research is dedicated to the detection of filtered microorganisms in aquatic biotopes [6]. At the same time, there is practically no information about the diversity and physiology of this bacterial compartment in soils in general and in particular in such extreme biotopes as the soils of Antarctica.

In our previous work it was revealed that half or even more of the bacteria cells in Antarctic soil are represented by ultra-small cells [9]. Moreover, in Antarctic soils, an increased content of such small cells was noted in comparison with the soils of temperate latitudes [10]. This data raises several questions: does the FFB fraction in Antarctic soils consist more of ultramicrobacteria or are they dormant cells? What is the functional role of FFB in the adaptation of bacteria to extreme living conditions, and what is the significance of the increased content of FFB for the stability of the entire microbial community?

We started our research of FFB in Antarctic soils by studying the soils from the Larsemann Hills. The number of FFB in these samples varied from 86 to 260 million cells per 1 g of soil, the relative content of the total number of bacterial cells was 20–90%. The greatest number and percentage of FFB were observed in samples from mineral and subsurface horizons. Analysis of 16S rRNA gene clone libraries showed that all filtered bacteria belong to the phylum *Proteobacteria* [9,11]. In this paper, we continued to study the FFB fraction in soil samples from the Bunger Hills (East Antarctica), which is quite inaccessible and therefore poorly studied, and fully fledged soil and microbiological studies have only recently begun there. This is a unique place on the planet in terms of studying the least affected ecosystems. Due to the peculiarities associated with the selection and transportation of soil samples from such a remote region, it is necessary to find effective techniques and methods that would help to obtain maximum information about the number, taxonomic diversity and viability of microbial cells in soils. We assumed that analysis of bacterial succession will allow us to trace the dynamics of the abundance and taxonomic diversity of bacteria and their microforms in the conditions that can develop during the warm periods of the year in Antarctica, as well as to most fully characterize the diversity of soil bacteria.

The aim of this study was to determine the total abundance and taxonomic diversity of bacteria and FFB during succession initiated by increased humidity and positive temperatures in soil samples from the Bunger Hills.

## 2. Materials and Methods

### 2.1. Materials

The objects of this study were soil samples taken and classified in 2018 during the Russian Antarctic expeditions in the coastal part of East Antarctica on the territory of the Bunger Hills (eastern part of Queen Mary Land, the field base “Bunger Hills” 66°17′ S 100°47′ E (Figure 1)). This is one of the largest ice-free area of Antarctica after the Dry Valleys, there is a wide variety of landscape conditions, with many small and large reservoirs filled with melt water in summer. From the moment of sampling to the beginning of our research, the samples were stored in a freezer at a temperature of −18 °C.

We selected three soil samples, in which a large number of FFB cells were previously identified [11], to initiate bacterial succession: a sample of Typic Haploturbel under a soda crust, № BB 63–48 (depth 10–15 cm) and 2 samples from the AT and AB horizons of Typic Haplorthel № BB 63–58, on the surface of which the development of a moss cover was observed (depth 0–2 cm and 10–15 cm, respectively) (Table 1).

### 2.2. Methods

#### 2.2.1. Physico-Chemical Analyses of Samples

The total content of carbon (C) and nitrogen (N) was determined using the method of dry combustion in a stream of oxygen on an Elementar Vario EL III elemental analyzer (Germany). The indicators were measured in triplicate.

Optical microscopy and scanning electron microscopy (SEM) were employed to describe morphologies and reveal occurrences of organo-mineral assemblages in samples at meso- and microscale. Mesomorphology of samples was studied in reflected light under Leica MZ6 StereoZoom microscope (Leica Microsystems, Wetzlar, Germany) equipped with the digital camera. Micromorphology of individual loci was examined under SEM (JEOL JSM-6610LV, Japan) in secondary electrons mode. For elemental analyses, SEM was combined with the energy-dispersive X-ray spectroscopy (EDX detector, Oxford Instruments, UK).

#### 2.2.2. Scheme of the Experiment to Initiate Succession

Each soil sample weighing 30 g was placed in a sterile Petri dish, moistened to 30%, mixed thoroughly, divided into two parts, each of which was placed in a desiccator. One desiccator was incubated at 5 °C, and the second at 20 °C. Sampling was carried out on days zero, 4, 7, 14, 30, 60, 90 and 120. At each point of the succession experiment, 2 g of soil was sampled. Each time in the samples, the number of total cells and the number of filterable forms were determined, and total genomic DNA was isolated from the soil and from the FFB fraction for further next-generation sequencing. In addition, at each point in the sequence, the soil suspension was plated on agar medium (TSA, R-2A).

#### 2.2.3. Obtaining the Fraction of Filterable Forms of Bacteria

Water soil suspension (1:100) was sonicated on an UZDN-1 disperser (INLAB, Russia) for 2 min at 0.44 A, 15 kHz to separate bacteria from soil particles and filtered through membrane filters (GP Millipore Express PLUS Membrane) with a pore size of 0.2 μm. The resulting filtrate was centrifuged (10 min, 10,000× *g*) to concentrate the cells, and the precipitated cells were used to isolate total DNA.

#### 2.2.4. Determination of the Total Number of Bacteria and FFB

The total number of bacteria and the number of the FFB were determined using fluorescence microscopy using the dye acridine orange. For this, the soil suspension or filtrate containing the FFB treated as described above was applied to defatted glass slides in a volume of 0.01 mL and spread over surface of 4 cm². Then the preparation was dried in air and fixed over a burner flame. Staining with acridine orange (1:10,000, biotium) was carried out for 2–3 min and then washed with distilled water. For each sample, 3 preparations were prepared. In each preparation, 90 microscope fields were viewed. The calculation of the number of cells was made according to the formula: N = S_1_ × a × n/V × S_2_ × C, where N—The number of cells in 1 g of soil; S_1_—Surface of the preparation (µm²); a—The number of cells in one field of microscope (average for all preparations); n—The dilution index of the soil suspension (mL); V—The volume of the drop applied to the glass (mL); S_2_—The surface of the field of view of the microscope (µm²); C—Aliquot (1 mL). The number of bacterial cells was counted using a ZEISS Axioscope 2+ fluorescence microscope (Carl Zeiss, Wetzlar, Germany).

#### 2.2.5. Isolation of DNA

Total DNA was isolated from soil samples and the FFB fraction using the PowerSoil DNA Isolation Kit (MO BIO, Carlsbad, CA, USA), following the manufacturer’s instructions.

Genomic DNA from the cultivated bacterial cells was isolated using the GeneJET Genomic DNA Purification Kit” (Thermo Scientific, Vilnius, Lithuania), following the manufacturer’s recommendations.

#### 2.2.6. Next-Generation Sequencing of the 16S rRNA Gene from Total Soil DNA

This method was used to analyze the structure of microbial communities in the studied samples. Library preparation and sequencing were done in the BioSpark laboratory (Russia). DNA libraries for sequencing were prepared using the two-stage PCR method. At the first stage, the hypervariable region V3–V4 of the 16S rRNA gene was amplified using primers Pro341F (5′-CCTACGGGNBGCASCAG-3′) and Pro806R (5′-GGACTACHVGGGTWTCTAAT-3′) [12]. At the second stage, the PCR product was amplified for bar-coding the library. DNA libraries were analyzed by the method of paired-end reads (2 × 300 bp) via generation of at least 10,000 paired reads per sample with Illumina MiSeq (USA). The sequencing data were processed with QIIME [13]. The algorithm with an open-reference classification threshold of 97% was used to divide the sequences into operational taxonomic units (OTU). The 16S rRNA sequence alignment of the reads and sequence distribution between taxonomic units were performed using Silva v. 132 [14].

#### 2.2.7. Amplification and Analysis of the 16S rRNA Genes from Bacterial Isolates Grown on Nutrient Media

The 16S rRNA gene fragments were amplified using primers 63F (5′-CAGGCCTAA CACATGCAAGTC-3′) and 1387R (5′-GGGCGGWGTGTACAAGGC-3′) [15]. PCR was performed as follows: 95 °C, 4 min; then 95 °C, 1 min; 55 °C, 1 min; 72 °C, 1 min 30 s (30 cycles); and 72 °C, 5 min. PCR products were purified on a columns using a GeneJET PCR Purification Kit (Thermo Scientific, Vilnius, Lithuania). DNA sequencing was performed using a set of reagents (BigDye ™ Terminator v. 3.1, Applied Biosystems, USA) with the subsequent analysis of the reaction products on an automated sequencing system (Applied Biosystems 3730 DNA Analyzer, USA) at the Genome Center for Collective Use (Institute of Molecular Biology, Russian Academy of Sciences, Moscow, Russia). The 16S rRNA gene sequences were analyzed online with BLAST [16]. Phylogenetic positions were determined using the data from NCBI (ncbi.nlm.nih.gov; accessed on 18 October 2020).

#### 2.2.8. Statistical Data Processing Methods

Box-and-whiskers diagrams were constructed using Microsoft Excel for statistical processing of the data obtained using direct fluorescence microscopy.

The diversity of Antarctic bacterial communities was assessed by the indices calculated using the number of sequences. Alpha diversity was assessed by Shannon Index (number 2 was taken as the logarithm base) and Simpson Index. Normalization of samples when calculating diversity indices was carried out for 9244 sequences (the minimum number of obtained sequences per sample). The analysis of community similarity (beta diversity) was carried out using the UniFrac method.

#### 2.2.9. Depositing the Received Data

The 16S rRNA gene sequences from total soil DNA obtained by next-generation sequencing have been deposited with the NCBI Sequence Read Archive (SRA) and are available under accession number PRJNA609617.

## 3. Results

### 3.1. Morphology and Composition of Soil Samples

#### 3.1.1. Mesomorphology

The material of AT horizon (BB 63–58) consisted of a dead moss tissue, partially decomposed organic matter (peat) and numerous mineral grains (Figure 2a) produced by weathering of the surrounding gneiss rock and entrapped later in the organic matrix of AT horizon. Except for a small portion of angular quartz debris, most of the mineral particles were covered by thin brownish organo-mineral coatings (Figure 2b). The underlying AB horizon had explicitly lower content of moss residues but much higher content of silt/clay fine earth that sprinkled the surface of primary mineral grains, which were represented mostly by mm-sized quartz, feldspars, pyroxenes and amphiboles (Figure 2c,d). In the Typic Haploturbel sample (BB 63–48), the organo-mineral material was organized in mm-sized granules that often had a coarse mineral grain in their core and a mixture of clays, carbonates and other salts at a periphery; the latter were mainly responsible for a lighter color in comparison to the other two samples (Figure 2e,f).

#### 3.1.2. Micromorphology

The SEM analysis suggests that most of bacteria were entrapped within complex organo-mineral films and co-occurred with amorphous carbonaceous material and clay minerals (Figure 2g–i and Figure 3). In the three studied samples we found no detached bacterial morphologies and surprisingly no cyanobacteria-dominated EPS (extracellular polymeric substances)—Rich biofilms that usually occur in East Antarctica topsoil and exhibit vivid structures at a microscale. The most frequently observed pattern related to organo-mineral associations was represented by numerous angular clay-sized domains and clusters with sharp edges (proxy of clays) embedded in an amorphous carbonaceous substance (proxy of EPS) with a few cell-like morphologies distinguished within this organo-mineral matrix. In samples from BB 63-58 soil profile, we also observed the clusters of mycelia (Figure 2h,i) with degraded spore-like structures, which likely correspond to *Actinobacteria* sp. [17].

The distribution of organo-mineral associations was tightly linked to the roughness of mineral surfaces with the bulk of them found in microdepressions (~100 µm wide and ~10 µm deep). Spatial distribution of chemical elements occurrences obtained from energy-dispersive X-ray spectroscopy indicated that such microdepressions are filled with clays in close association with organogenic substances including those exhibiting bacteria-like morphologies. Figure 3 demonstrates the co-occurrence of carbonaceous material (C as a proxy) and clay particles (Al and Si as a proxy) in microdepression on quartz (Si and O as a proxy).

### 3.2. Dynamics of the Total Number of Bacteria and FFB in the Studied Soils

#### 3.2.1. Dynamics of the Total Number of Bacteria in the Studied Soil Samples

In the sample of Typic Haploturbel (BB 63–48), there were slight fluctuations in the total number of live bacteria, which varied within the limits of 0.4–20 × 10^8^ cells per 1 g of soil at 20 °C during the incubation period. The maximum number was observed at the initial point of the succession experiment, then the total number of bacteria fell until the end of succession with a small peak on day 7 (Figure 4I). The population dynamics during soil incubation at 5 °C was similar, only the maximum number was observed on day 14 (Appendix A).

In the samples of Typic Haplorthel (BB 63–58 AT and BB 63–58 A B), the total number of bacterial cells at the initial point of succession was higher than in the sample of Typic Haploturbel. The dynamics of the total bacterial count during the succession experiment fluctuated in both samples. The maximum number of bacteria was observed at three points: the initial, on the 30th and 90th day. The minimum number of bacteria was observed on days 7, 14, and 60. Graphs of population dynamics at a temperature of 5 °C are presented in the Appendix A.

Such wave-like character of the dynamics of the total number of bacterial cells may be associated with a lack of nutrients in the samples. Cells are reactivated with an increase in the soil moisture content and in temperature, but go into a dormant state due to the depletion of available sources of N and C, then cells that can use the other organic substances are activated (Figure 4I).

#### 3.2.2. Dynamics of FFB during Succession

The maximum number of FFB in samples BB 63–48 and BB 63–58 AB was observed at the initial point of succession at 20 °C (16–22 × 10^8^ cells per 1 g of soil). Then this number decreased at the control point for 3 days (up to 5 × 10^8^ cells per 1 g of soil), slightly increased at days 7 and 14 (10–15 × 10^8^ cells per 1 g of soil), and then remained approximately at the same level until the end of succession (about 5 × 10^8^ cells per 1 g of soil) (Figure 4II). In the sample BB 63–58 AT the maximum peak in the number of FFB was observed not in the initial samples, but on the 14th day of succession (20 × 10^8^ cells per 1 g of soil), declines in the number of FFB cells were registered on the 3rd and 30th days (up to 5 × 10^8^ cells per 1 g of soil) (Figure 4II). The minimum number in all three samples was observed on the 3rd and 30th days of succession, i.e., when the total number of bacteria reached the maximum. Graphs of the dynamics of the number of FFB at 5 °C are presented in the Supplement (Appendix A).

After the end of the experiment (120 days), the total content of C, N was also determined in the soil samples using the dry combustion method, as at the beginning of succession. No significant changes in the content of these elements were found after the succession initiated by positive temperatures, which serves as an additional evidence of the stability of the functioning of bacterial communities (Table 2).

### 3.3. Taxonomic Diversity of Communities in the Studied Samples

During the succession experiment, taxonomic diversity was determined using next-generation sequencing at 0, 14, 60, and 120 days. The analysis was carried out both in the total DNA isolated from soil and DNA from the fraction of filtered cells. This made it possible to monitor the changes that occurred during succession at the level of the taxonomic structure of soil microbial communities. Unfortunately, we were able to isolate not much DNA from the samples during the study. The concentration of isolated DNA was 7–34 ng/uL for soil samples and 5–21 ng/uL (NanoDrop Spectrophotometer, Thermo Scientific) for FFB fraction.

#### 3.3.1. Taxonomic Diversity of Bacteria in the Studied Soils at the Phylum Level

The distribution of phylum ratios in the samples remained virtually unchanged during succession and did not depend on the incubation temperature of the soil. In the sample BB 63-58 AT, the dominant phyla were *Chloroflexi* (28–32%), *Proteobacteria* (18–25%), and *Acidobacteria* (17–21%) (Figure 5A1,A2). In the sample BB 63-58 AB, the dominant phyla were *Chloroflexi* (16–37%), *Proteobacteria* (20–27%), and *Acidobacteria* (15–21%) (Figure 5B1,B2). In the sample of BB 63–58 AB, a higher content of representatives of the phylum *Firmicutes* (0.5–13%) was observed compared to the sample of BB 63–58 AT. Sequencing of total DNA isolated from samples of the soil sample BB 63–48 failed at all points of succession, while for the FFB fraction the sequencing was successful only for samples taken at 60 and 120 days (Appendix A).

#### 3.3.2. Taxonomic Diversity of FFB in the Studied Soils at the Phylum Level

It was found that the same phylogenetic groups are represented in the fraction of FFB cells as in the entire soil community. Only the ratio of the dominant phyla was different. In the sample BB 63–58 AT, such phyla as *Proteobacteria* (37–84%), *Chloroflexi* (7–30%) and *Actinobacteria* (0.5–12%) dominated (Figure 5A1,A2). During succession, the FFB fraction was increasingly enriched with representatives of the phylum *Proteobacteria*. In this sample, the influence of the incubation temperature of the soil was stronger: the proportion of the phylum *Proteobacteria* was greater at the incubation temperature of 5 °C, and the proportion of the phyla *Chloroflexi* and *Actinobacteria* was greater at 20°.

In the sample BB 63–58 AB, the dominant phyla were *Proteobacteria* (70–81%), *Actinobacteria* (7–12%), and *Chloroflexi* (3–7%) (Figure 5B1,B2). In this sample, the distribution of phyla remained virtually unchanged during the succession experiment and did not depend on the incubation temperature of the soil.

In the sample BB 63–48, more than 90% of the bacteria belonged to the phylum *Proteobacteria*. The phyla *Actinobacteria* (1–5%), *Firmicutes* (0.06–4%), and *Bacteroidetes* (0.6–3%) were also found (Appendix A).

Such phyla as *Acidobacteria*, *Cyanobacteria*, *Saccharibacteria*, *Gemmatimonadetes* were detected in the fraction of FFB cells during succession, although they were not observed in the initial samples.

#### 3.3.3. Taxonomic Diversity of Bacteria in the Studied Soils at the Genus Level

Figure 6 and Figure 7 shows the groups of OTUs or genera, if the latter could be classified, that make the most significant contribution to the structure of the community of samples from the profile BB 63–58. Within a single soil sample, the composition and the ratio of dominant groups (or genera) do not change during succession and do not depend on temperature.

Among the dominant groups in both samples of the profile BB 63–58, representatives of the class *Ktedonobacterales* (phylum *Chloroflexi*) attract attention. This group of bacteria is interesting because the cultured *Ktedonobacteria* are Gram-positive aerobes and have a complex morphology and differentiation, form a branched mycelium with spores, such as the mycelium of actinomycetes, as a result of which this group is predicted to produce various secondary metabolites. In addition, as shown by previous studies based on the analysis of clone libraries or on next-generation sequencing of the 16S rRNA gene of bacteria, sequences related to *Ktedonobacteria* were found in various extreme biotopes, such as volcanic, Antarctic and cave ecosystems [18]. Thus, representatives of this group of microorganisms are most likely typical inhabitants of Antarctic soils. In the FFB fraction, representatives of *Ktedonobacteria* are also found among the dominant OTUs, namely Uncultured *Ktedonobacterales* (JG30a-KF-32) and unclassified Uncultured OTU *Ktedonobacterales* (Figure 6 and Figure 7).

Another interesting group whose representatives are found in the studied samples is the phylum *Gemmatimonadetes*. Representatives of *Gemmatimonadetes* are often found in the 16S rRNA gene libraries of bacteria isolated from various natural environments, and are considered one of the nine main phyla found in soils, which accounts for about 2% of the total soil bacterial community [19]. The largest proportion of representatives of *Gemmatimonadetes* is found in arid soils, which suggests the adaptation of this group of bacteria to habitats with low humidity [20]. The genus *Gemmatimonas* is one of the dominant ones in the sample from the horizon AT, but in the horizon AB this genus is not among the dominant groups, which is probably due to the difference in the moisture content in the horizons (Table 1).

In the FFB cell fraction, representatives of the genus *Gemmatimonas* are also not among the dominant OTUs (Figure 6. Instead of the genus *Gemmatimonas*, an uncultivated representative of the order *Halanaerobiales* (ODP 1230B8.23), belonging to the phylum *Firmicutes*, was the most common in the sample from the horizon AB. However, on the 120th day of the succession experiment, this order also came out of the group of dominant OTUs, which may indicate a change in soil conditions during succession (Figure 7).

The horizon AT, in comparison with AB, contains more organic matter (Table 1). Perhaps this is why an unclassified representative of the family *Chitinophagaceae* (phylum *Bacteroidetes*) is present among the dominant OTUs in this horizon, since the representatives of this family are aerobic heterotrophic microorganisms capable of decomposing various organic substances.

Among the dominant groups of bacteria in the soil samples, representatives of the phylum *Acidobacteria* are particularly distinguished, including ambiguous taxa Subgroup 2, *Bryobacter*, uncultured candidatus *Solibacter* and uncultured *Acidobacteriaceae* (Subgroup 1). At the same time, in the FFB fraction, these taxonomic groups are not dominant. The exception is the representatives of the genus *Bryobacter*. However, even those were found only on the 14th day of succession in a single sample from the horizon AT (Figure 6).

In the soil samples, among the dominants, there are representatives of uncultured *Chthoniobacterales* (Candidatus *Udaeobacter* clone DA101) belonging to the phylum *Verrucomicrobium*, which was previously found in the pasture soil. However, in the FFB fraction, this OTU does not belong to the dominant group. Representatives of the order *Chthoniobacterales* are common inhabitants of soils, but have not yet been sufficiently studied. According to genomic data, *Spirobacteria* (*Chthoniobacterales*) promote carbon cycling by decomposing various complex carbohydrates, such as cellulose and xylan [21].

Noteworthy that representatives of some genera of Proteobacteria dominate only in the FFB fraction, but not in soil samples. Thus, representatives of the genera *Methylobacterium*, *Pseudomonas*, and *Sphingomonas* dominate in the FFB fractions, and other representatives of Proteobacteria are more common in soils, including ambiguous taxa (SC-I-84), *Bradyrhizobium*, uncultured *Caulobacteraceae*.

Since the phylum *Actinobacteria* was one of the dominant ones, the contribution of its individual representatives was significant. Such OTUs as *Acidothermus*, uncultured *Acidimicrobiales* were found both in soils and in the FFB fraction. However, for example, representatives of the order *Solirubrobacterales* (class *Thermoleophilia*) were not found in the FFB fraction, but only in soils. It is noteworthy that the sequences of organisms belonging to the class *Thermoleophilia* are constantly found in various Antarctic samples, sometimes reaching a content of 15% [22,23].

Since no sequencing data could be obtained for the soil sample from the BB 63–48 profile, the dominant genera in it were analyzed only for the FFB fraction. It showed a high content of representatives of the phylum *Actinobacteria* (genera *Arthrobacter* and *Rhodococcus*), *Firmicutes* (genus *Bacillus*) and *Proteobacteria* (genera *Bdellovibrio*, *Methylobacterium*, *Pseudomonas*, *Sphingomonas*, uncultured *Escherichia-Shigella*) (Appendix A).

Thus, both microbial soil communities and the FFB fraction have their own spectrum of dominant OTUs. In addition, if at the level of phyla, the similarity between them is clearly traced, then at the level of orders and genera the difference becomes noticeable. Not all OTUs dominant in the soil were also dominant in the FFB fraction, and vice versa.

### 3.4. Analysis of Diversity Indices

Based on the data obtained by next-generation sequencing, we calculated the Simpson and Shannon diversity indices. The values of diversity indices both in the soil and in the FFB fraction changed slightly during succession. For example, the average value of the Simpson Index for the sample BB 63–58 AT at 5 °C was 0.967 ± 0.005 for soil and 0.930 ± 0.019 for the filtrate (Table 3), and the Shannon Index was 6.5 ± 0.2 and 5.3 ± 0.4, respectively (Table 4). For the sample BB 63–58 AB at 20 °C, the average value of the Simpson Index was 0.966 ± 0.012 for the soil and 0.936 ± 0.018 for the filtrate (Table 3), while the Shannon Index was 6.3 ± 0.4 and 5.6 ± 0.2, respectively (Table 4).

It should be noted that the diversity indices for the FFB fraction were slightly lower than for the entire community. The lowest diversity indices were found in Typic Haploturbel. For soils, during succession, the value of the Shannon Index ranged from 5.3 to 6.7, and the most diverse was the sample of BB 63–58 AT at 120 days of succession. The alpha-diversity indices of the microbial community of the Bunger Hills soils were slightly higher than the diversity indices calculated for other East Antarctic soils, so the average values of the Shannon Index in samples from the Larsemann Hills ranged from 4.02 to 5.31, and the Simpson Index varied less significantly among the studied samples from 0.945 in the transition horizon to 0.985 in the upper layer of soil from the anthropogenic-contaminated site of Progress station [24]. In another study, the average values of the Shannon Index in samples from various areas of the leading edge of the glacier, also located in the Larsemann Hills, ranged from 3.15 to 4.78, and the values of the Simpson Index from 0.0082 to 0.0505, although it is likely that the authors used the inverse Simpson Index: 1-C (where C is the Simpson Index), the so-called “probability of interspecific encounters” [25].

### 3.5. Analysis of Phylogenetic Similarity of Bacterial Communities

Figure 8 shows the results of the unweighted UniFrac analysis in the form of a two-dimensional diagram. The close location of soil samples BB 63–58 AT and BB 63–58 AB in the diagram indicates that the taxonomic structures of their bacterial communities are similar. On the contrary, the microbiomes from the FFB fraction form a separate, more scattered group, which means that their bacterial communities are more different from each other and from the fraction of cells of normal size.

No significant changes in taxonomic diversity were observed in the total fraction of bacteria isolated from soil samples during succession, in contrast to the FFB fraction. Consequently, the FFB fraction is a more dynamic, more rapidly changing cell fraction compared to the entire soil microbiome. Apparently, this is because a part of the cells constantly goes into a dormant state and comes out of it, depending on the prevailing environmental conditions.

We assume that monitoring of the FFB fraction in the course of changing conditions (during succession) helps to better understanding the processes occurring in bacterial communities, and allows us to more fully characterize the taxonomic diversity of the community. Table 1 in the Supplement provides a list of unique OTUs represented either only in soil samples, or only in the FFB fraction. In particular, representatives of the class *Alphaproteobacteria* (*Ochrobactrum*, *Roseomonas*, *Paracoccus*, *Afipia*), the class *Betaproteobacteria* (*Achromobacter*, *Polarmonas*, *Delftia*, *Ralstonia*, *Methylophilaceae*, *Hydrogenophilaceae*), the phylum *Actinobacteria* (*Blastococcus*, *Geodermatophilus*, *Arthrobacter*, *Micrococcus*, *Agroccus*, *Rhodococcus*, *Kocuria*, uncultured *Frankiales*), the phylum *Bacteroidetes* (*Chryseobacterium*, *Flavobacterium*, *Rufibacter*, *Hymenobacter*, *Pedobacter*) were found only in the FFB fraction (Appendix A). Perhaps this is because these taxa have a low representation in the fraction of cells of normal size and, therefore, it is difficult to isolate the optimal amount of DNA for further sequencing.

### 3.6. Alphaproteobacteria/Acidobacteria Ratio

We evaluated the ratio of *Alphaproteobacteria* to *Acidobacteria*, as each of these phyla plays its own important role in soil bacterial communities. Representatives of the phylum *Alphaproteobacteria* are among the fastest growing bacteria, preferring nutrient-rich environments. On the contrary, representatives of the phylum *Acidobacteria* are characterized by slow growth and prefer soils with a low nutrient content. Thus, a high value of the ratio of *Alphaproteobacteria* to *Acidobacteria* indicates a large supply of nutrients in the soil and an increase in CO_2_ production from the soil [26,27].

The calculated ratios of *Alphaproteobacteria* to *Acidobacteria* in the studied soil samples of the Bunger Hills are presented in Table 5. The small value of the ratio of *Alphaproteobacteria* to *Acidobacteria* in the soils confirms that Antarctic soils do not have a high content of nutrients and representatives of the phylum *Acidobacteria* play a significant role in the bacterial community. The increased ratios of *Alphaproteobacteria* to *Acidobacteria* in the fraction of filtered cells is noteworthy, while in the soil this ratio practically did not change over time and averaged 0.72 ± 0.08 for the sample BB 63–58 AT, and 0.89 ± 0.24 for the sample BB 63–58 AB. In the fraction of filtered bacterial forms, the ratio of *Alphaproteobacteria* to *Acidobacteria* increased significantly during the succession experiment. For example, in the sample BB 63–58 AT at 5 °C, it changed from 12.5 at the beginning of succession to 130 at the end (Table 5). This may indicate that under favorable conditions of growth, the available nutrients were absorbed and, as a result of starvation, the representatives of *Alphaproteobacteria* came into a dormant state.

### 3.7. Number and Taxonomic Diversity of Cultured Bacterial Cells

The number of CFUs (colony-forming unit) in 1 g of soil varied during the course of succession in the studied samples, but in general it was rather low. In particular, it did not exceed 1500 × 10^3^ CFUs per 1 g of soil for 60 days of succession in the sample BB 63–58 AB. Not a single CFU grew from the sample BB 63–48 at the incubation temperature of 5 °C throughout the entire succession, and colonies began to appear only from the 3rd day of succession at 20°C. A larger number of cultivated bacteria was observed on the R-2A medium than on the TSA medium for the samples from the profile BB 63–58. The number of heterotrophic bacteria constantly increased during the succession experiment, except for sample AB during soil incubation at 5 °C. The number of bacteria ranged from 1 × 10^3^–950 × 10^3^ CFUs per 1 g of soil in sample AT to 6 × 10^3^–1500 × 10^3^ CFUs per 1 g of soil in sample AB (Appendix A). The obtained data on the number of heterotrophic organisms are comparable with the data obtained for samples from other East Antarctica oases. So, the number of cultivated heterotrophs in samples from the Larsemann Hills ranged from 2.0 × 10^3^ to 2.8 × 10^5^ CFUs per 1 g of dry soil in the glacial transect to 5.0 × 10^2^ and 1.0 × 10^8^ CFUs per 1 g of dry soil from the periglacial part of the Hills [25].

We didn’t find diverse heterotrophic bacterial complex when the soil filtrate was inoculated on TSA and R-2A nutrient media; representatives of the genus *Pseudomonas* dominated at all stages of succession. 13 strains of Gram-negative bacteria and 14 g-positive were isolated from the studied samples. Even though bacteria of the genus *Pseudomonas* dominated throughout the succession experiment, more diverse strains of Gram-positive bacteria were isolated at the initial stages, and Gram-negative bacteria dominated in the later stages (Appendix A).

## 4. Discussion

Considering distributions of Na, K, Ca, Mg, Fe we suggest here an intimate interplay of clays, bacteria and EPS that constitute complex organo-mineral films present in microcavities on primary minerals. It is likely that a significant portion of bacteria in the studied samples co-occurred with clay particles or was even enveloped by flakes of clays. Such microhabitats also known as “clay hutches” could provide an effective survival unit for bacteria with a nutritional pool satisfying their minimal requirements [28]. It is very common for subaerial bacteria and other microorganisms to form a biofilm with an abundant hydrated EPS matrix that helps to tolerate various extremes on the mineral surfaces. Incorporation of clay minerals into biofilm (formation of organo-mineral film) can provide additional protection and physical support, while clays, in general, have a tremendous effect on microbial growth, survival, nutrients acquisition, respiration, biosynthesis and antioxidant system [29]. Serving as a source of nutrients or a sink for the microbial waste products clay minerals may not only alter microbial metabolism but also promote the growth of taxonomically different bacteria [29].

In comparison to primary minerals, clays provide an enormous pool of surfaces and geochemical niches for bacteria, and this could become particularly important in a weakly weathered environment of East Antarctica where clay content is low. Small dormant forms, as well as ultramicrobacteria, might need surfaces with high sorption capacity to retain in soil and it is possible that clays serve as an important habitat for them as well as for the bacteria of a common size. It is of particular interest if clays-EPS assemblages serve as refugia for small dormant forms and/or ultramicrobacteria (especially considering the submicron size of clay domains) that help to overcome a period of extreme conditions and resume activity once the setting is more favorable.

The taxonomic diversity at the phylum level found in the Antarctic samples studied by us does not differ from that in other soils of East Antarctica. During analysis of soil samples from the Larsemann Hills and polar desert of the Browning Peninsula representatives of the phyla *Proteobacteria, Acidobacteria, Actinobacteria*, and *Chloroflexi* dominate [23,24,25]. It should be noted that in all these studies were revealed that representatives of the phylum *Cyanobacteria* play a significant role in the bacterial communities, and in the samples from the Bunger Hills studied by us, this phylum was a minor component of the community. Apparently, the content of cyanobacteria may depend on the season and the presence of melt water in the soil, but this hypothesis requires further study.

The results of our study showed that the structure of the taxonomic community at the phyla level and at low taxonomic levels (orders/classes/genus) does not change dramatically during a 120 days succession under the specified experimental conditions (temperature 5 °C and 20 °C, humidity up to 30%). Perhaps, this is because the restructuring of the community requires much more time, as shown in the work of Yergeau and coauthors. This study investigated the effects of warming on various subantarctic (Falkland Islands) and Antarctic (Signy and Anchorage Islands) soils over 3 years using open-top chambers (OTCs) and showed significant increases in fungi and bacteria and the ratio of α-*Proteobacteria*/*Acidobacteria*. However, the authors also did not observe strong changes in the general structure of the soil community at low taxonomic levels (genus/species level) after warming; however, warming caused significant changes in the composition of the bacterial community at higher taxonomic levels (type/class) [30].

It is also possible that significant shifts in the microbial community of Antarctic soils require an increase in the availability of nutrients in the soil, in addition to an increase in temperature and humidity. A study conducted by Newsham and coauthors showed that changes in the structure of Antarctic bacterial communities are more influenced by the addition of nutrients than by changes in temperature. Thus, the addition of TSB dry nutrient medium to the soil (the experiment lasted 49 months) had a significant effect on the composition of the soil bacterial community, while an increase in the average monthly soil temperature to 2.3 °C did not show an obvious effect on the composition of the bacterial community of the Antarctic soil. In particular, the addition of TSB resulted in a very significant increase in the *Proteobacteria/Acidobacteria* ratio, and an increase in the content of *Firmicutes* and *Bacteroidetes*, while the abundance of most Gram-negative bacterial groups decreased (*Deltaproteobacteria*, *Chloroflexi*, *Acidobacteria*, *Cyanobacteria*, *Gemmatimonadetes*, *Planctomycetes* and *Verrucomicrobia*) [31]. Thus, it can be assumed that low humidity and the lack of nutrients have the main influence on the development of soil microorganisms in the Antarctic.

To answer the question whether the FFB fraction in Antarctic soils consists to a greater extent of ultramicrobacteria or dormant cells, it is necessary to analyze whether the already known representatives of obligate UMB are found in the filtrate obtained by us. In particular, the CPR (candidate phyla radiation) group is of a great interest. Even though most of the representatives of this group have not been cultivated so far, few cultivated bacteria of this group belonging to the “Ca. *Saccharibacteria*” (former TM7) have ultra-small cell size (200–300 nm) and a highly reduced genome, with a complete lack of the amino acid biosynthetic capacity [32]. In the samples analyzed in our study, OTUs belonging to the phylum *Saccharibacteria* were found; their content was low and did not exceed 0.3% in the soil and 4% in the FFB fraction. The highest content of OTUs related to this phylum was observed in the FFB fraction in the sample BB 63–58 AT (Appendix A).

Other important representatives of obligate UMB belonging to the Luna cluster (class *Actinobacteria*) [33] were not found in the studied samples. Representatives of the Luna cluster were previously found only in freshwater habitats and may not be typical representatives of soil microbial communities. In the FFB fraction in the Antarctic samples studied by us, there are many genera from the phylum *Actinobacteria* that were not previously found in the soil (Appendix A), which makes this phylum promising for the search for new undescribed UMB. In addition, we were able to isolate 15 strains of filterable forms of bacteria belonging to the phyla *Actinobacteria*, *Bacteroidetes*, *Deinococcus-Thermus*, *Firmicutes* and *Proteobacteria* in our previous study focused on the taxonomic diversity of bacteria and their filterable forms in the soils of the Larsemann Hills. Most (8 out of 15) isolated strains were assigned to the phylum *Actinobacteria* [11]. The strains obtained by us cannot be classified as obligate UMB, since the cells did not retain their ultra-small size when cultivated under conditions favorable for growth.

There is currently limited information on the UMB of soil habitats in contrast to aquatic habitats. Previously, Janssen et al. reported anaerobic obligate UMB isolated from soils of rice fields, assigned to the genus *Opitutus*, class *Verrucomicrobiales* with a cell volume of 0.030 μm^3^ [34]. OTUs belonged to the genus *Opitutus* were also found in both samples from profile BB 63-58 in the Antarctic soils studied by us. However, the OTUs data refer to minor components and their content in the soil does not exceed 0.04%. Representatives of the genus *Opitutus* were found only in the AT sample on the 14th day of succession in the FFB fraction (Appendix A). The absence of representatives of the genus *Opitutus* in the FFB fraction may be due to the insufficient amount of DNA required for detection using next-generation sequencing, due to the small number of cells in the filtrate.

In addition to UMB, slender filamentous bacteria are found in 0.2 µm filtrates from various environmental samples, which are supposed to “squeeze” through the pores. The shape of such cells is usually slender, filamentous, of variable length, but shows a pleomorphism with other shapes, such as spiral, spherical or curled. However, it has not yet been clarified whether each morphological form is associated with of dormant state [6]. Most of these cells isolated from the filtrates belong to the phylum *Proteobacteria**,* including *Hylemonella gracilis* [35]; *Oligoflexus tunisiensis* [36]; *Silvanigrella aquatica* [37]; *Ralstonia pickettii* [38]. Since our results have shown the enrichment of the FFB fraction of Antarctic soils precisely by the phylum *Proteobacteria*, it is possible to expect that a significant part of the cells are just those slender filamentous forms of bacteria capable of “squeezing” through the pores of the filter.

An increase in the ratio of *Alphaproteobacteria* to *Acidobacteria* in the FFB fraction was noted in our work during the succession experiment. Probably, during the initiation of succession, when the amount of available moisture increased, a significant part of the cells of *Alphaproteobacteria* revived, used the available nutrients from the soil, and then again passed into a dormant state, i.e., into the FFB fraction. On the other hand, representatives of *Acidobacteria* gradually disappeared from the FFB fraction in the course of succession and passed into the active form, as more suitable conditions for their existence came and ecological niches were vacated.

This assumption is also confirmed by the fact that among the dominant *Alphaproteobacteria* OTUs in the FFB fraction, a significant proportion were representatives of such groups as *Sphingomonas* (8–36%), *Methylobacterium* (6–26%), *Variibacter* (0.1–1%), *Methylocystacea* (0, 1–1%), while in the total bacterial fractions these groups were not dominant and their content did not exceed 0.6% (Figure 6 and Figure 7). On the contrary, several OTUs belonging to *Acidobacteria*, including uncultured bacterium candidatus *Solibacter*, *Bryobacter*, uncultured bacterium *Holophagae*, ambiguous taxa *Chitinophagaceae*, uncultured bacteria *Sphingobacteriales*, were not detected at the last stage of succession in the FFB fraction, while some increase in the content of these OTUs was observed in the total bacterial fraction in the last stages of succession (Figure 6 and Figure 7).

The enrichment of the FFB fraction in the genera *Sphingomonas*, *Methylobacterium*, and *Pseudomonas* may also be associated with a low level of contamination with microbial DNA during sample preparation. Potential sources of contamination include purified water, PCR-reagents and DNA extraction kits. Basically, contaminants are water- and soil-associated bacteria, including representatives of the genera *Methylobacterium*, *Pseudomonas* and *Sphingomonas*. The presence of contaminating DNA is a serious problem for samples containing low microbial biomass, an example of which is the FFB fraction [39]. The problem of identifying contaminant organisms for environmental samples such as soil or water is especially urgent, since it is almost impossible to distinguish them from the organisms actually present in these samples. To solve this problem, we tried to perform sequencing of negative control samples without the addition of bacterial fractions, treated with the same kits for DNA extraction and PCR amplification, but we could not receive enough DNA for sequencing.

At the same time, Lysak et al. revealed a high abundance of representatives of such groups of bacteria as *Actinobacteria*, *Cytophaga*, and *Proteobacteria* in soils by studying the distribution of cells passing through 0.2 μm filters. Using 16S rRNA-specific fluorescently labeled probes, it was shown that the dominant groups are Gram-negative bacteria (*Cytophaga* and *Proteobacteria*), which indicates a significant role of these phyla in the FFB fraction. It should be noted that in the above-mentioned work, the distribution of bacterial taxa in the soil (unfiltered soil suspension) was also approximately the same as among the FFB fraction [10].

The data obtained during the study on the dynamics of the number and taxonomic diversity of cells in the soil and FFB suggest that most of the bacterial cells in the filtrate are represented by starvation forms that can pass into the active state with normal size in the presence of favorable conditions for cell growth, while the content of obligate UMB in the filtrate is low. A similar conclusion was reached by the authors of a study examining three fractions of filtered seawater from the western Mediterranean Sea (Calvi, Corsica/France). It was determined that most of the filterable bacteria were starving forms of marine bacteria rather than ultramicrobacteria and belonged to the known typical marine isolates of the subclasses α- and γ-*Proteobacteria* [40].

Thus, the filtrate we obtained from the Antarctic soils likely mainly consists of cells capable of “squeezing” through the filter pores, as well as starving or dormant forms of bacteria, which may make a significant contribution to the stability of bacterial communities in Antarctic soils.

## 5. Conclusions

As far as we know, this work is the first study aimed at studying the functional role and viability of the ultra-small bacterial cell fraction in soils. The result of the experiment with the initiation of succession by different temperature regimes and additional moisture showed that the structure of the bacterial communities of the studied Antarctic soils is quite stable, since part of the cells of the microbial community is in a dormant state in the form of FFB and can quickly recover to active growth when favorable conditions occur and vice versa. Therefore, it can be assumed that FFB are a pool of cells that allows bacteria to survive in unfavorable environmental conditions: low temperatures, alternating freezing-thawing, lack of organic substrates, desiccation, etc. Probably, the rapid transition of cells of the active part of the bacterial population into small dormant forms is one of the strategies for survival in extreme conditions.

## Figures and Tables

**Figure 1 microorganisms-09-01728-f001:**
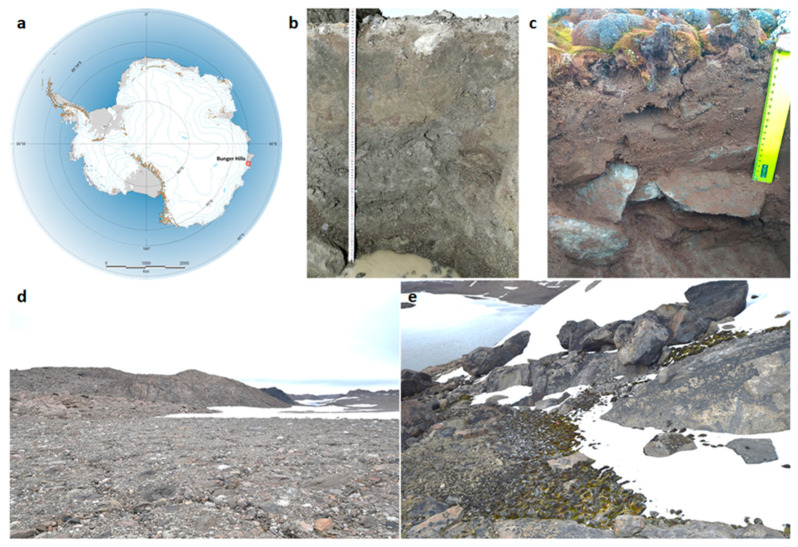
(**a**) Location of Bunger Hills, Eastern Antarctica (relief location map). Soil profiles: (**b**)—BB 63–48 (photo by I. Shorkunov), (**c**)—BB 63–58 (photo by A. Dolgikh) (see Table 1). (**d**)—Typical landscape of the central part of the Bunger Hills, flat valley, the location of the soil BB 63–48 (photo by I. Shorkunov). (**e**)—Typical wind shelters with moss cover in the southern part of the Bunger Hills, the location of the soil BB 63–58 (photo by I. Shorkunov).

**Figure 2 microorganisms-09-01728-f002:**
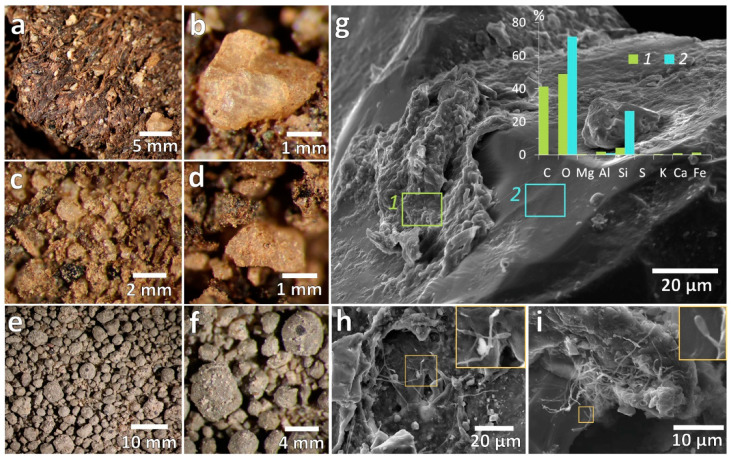
Morphology of intact soil samples under reflected light (**a**,**b**)—Horizon AT from BB 63-58 soil profile; (**c**,**d**)—Horizon AB from BB 63-58 soil profile; (**e**,**f**)—Sample from BB 63–48 soil profile) and Scheme 63. soil profile (**g**–**i**). Abundances of elements shown at (**g**) correspond to organo-mineral film (region of interest “1”) that covers quartz grain (region of interest “2”).

**Figure 3 microorganisms-09-01728-f003:**
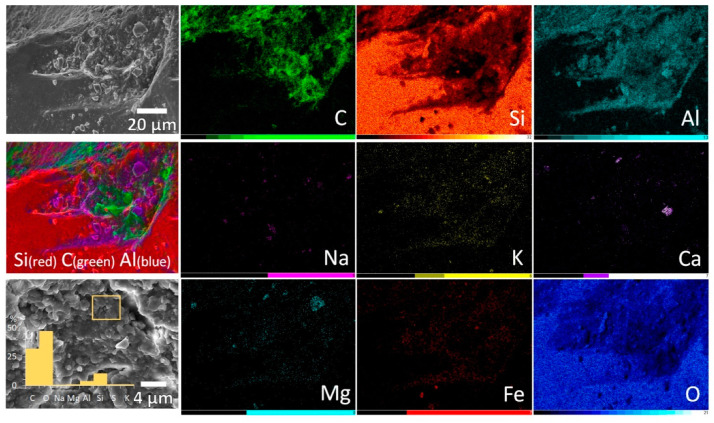
SEM micrographs and maps of elements occurrences obtained from energy-dispersive X-ray spectroscopy in a micron-scale cavity on quartz grain filled with clays in association with carbonaceous substances including bacteria (AB horizon from the BB 63–58 soil profile). The first left column from top to bottom: general view on the region of interest in secondary electrons; composite image indicating the close occurrence of carbon and clay particles (Al as a proxy) in microdepression on quartz (Si as a proxy); close view on organo-mineral film indicating the intimate interplay of clays, bacteria and EPS. All other columns—Maps of individual elements occurrences; the scale is the same as for the overview image.

**Figure 4 microorganisms-09-01728-f004:**
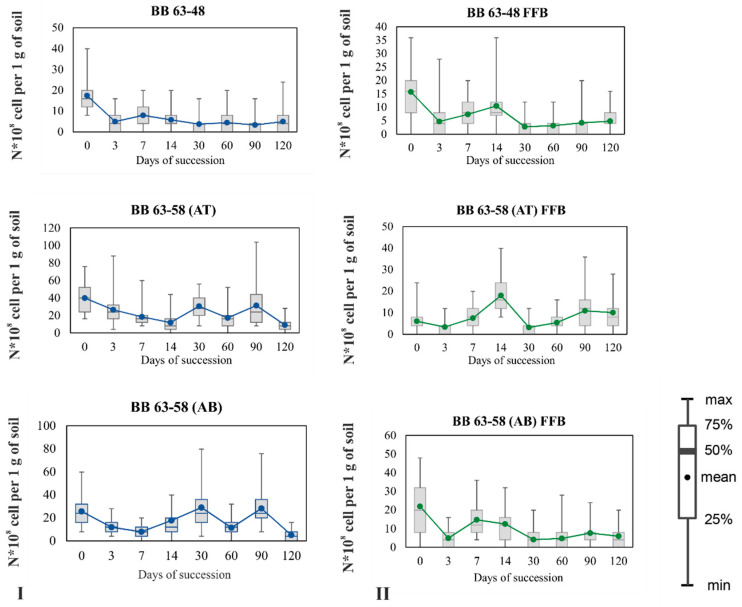
Dynamics of the total number of bacteria and FFB in the studied soils, incubation temperature 20 °C: (**I**)—Soil (total cells); (**II**)—FFB.

**Figure 5 microorganisms-09-01728-f005:**
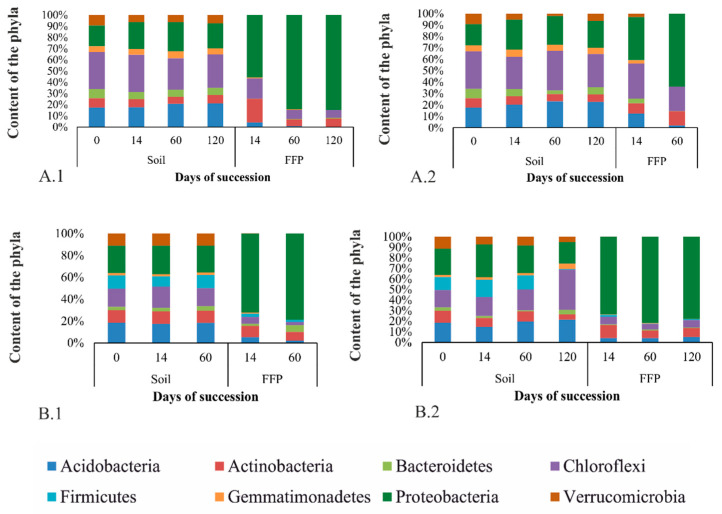
Distribution of bacterial phyla in the samples during succession (only phyla with content greater than 2% are presented). (**A1**)—BB 63–58 AT, 5 °C; (**A2**)—BB 63–58 AT, 20 °C; (**B1**)—BB 63–58 AB, 5 °C; (**B2**)—BB 63–58 AB, 20 °C.

**Figure 6 microorganisms-09-01728-f006:**
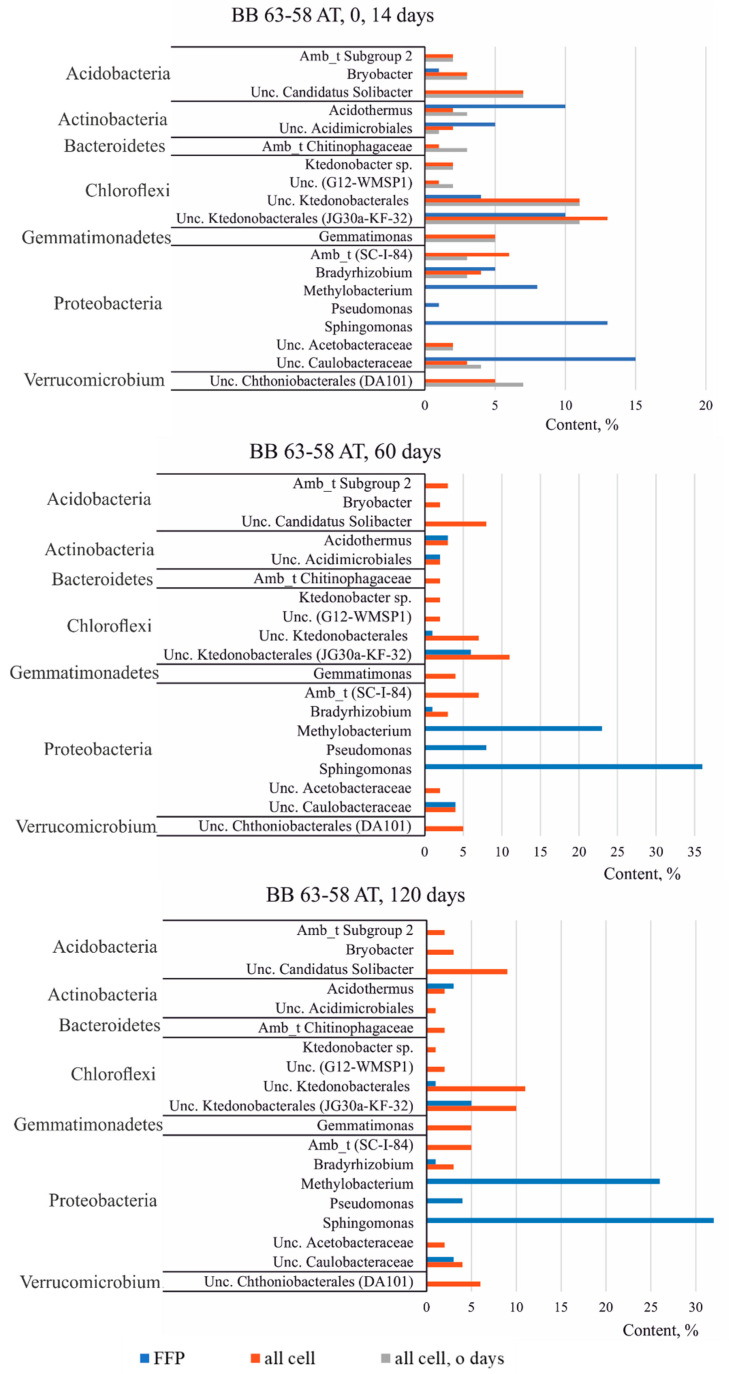
Distribution of dominant groups during succession in the horizon AT, BB 63–58 profile, 5 °C (phyla with content greater than 2% are presented).

**Figure 7 microorganisms-09-01728-f007:**
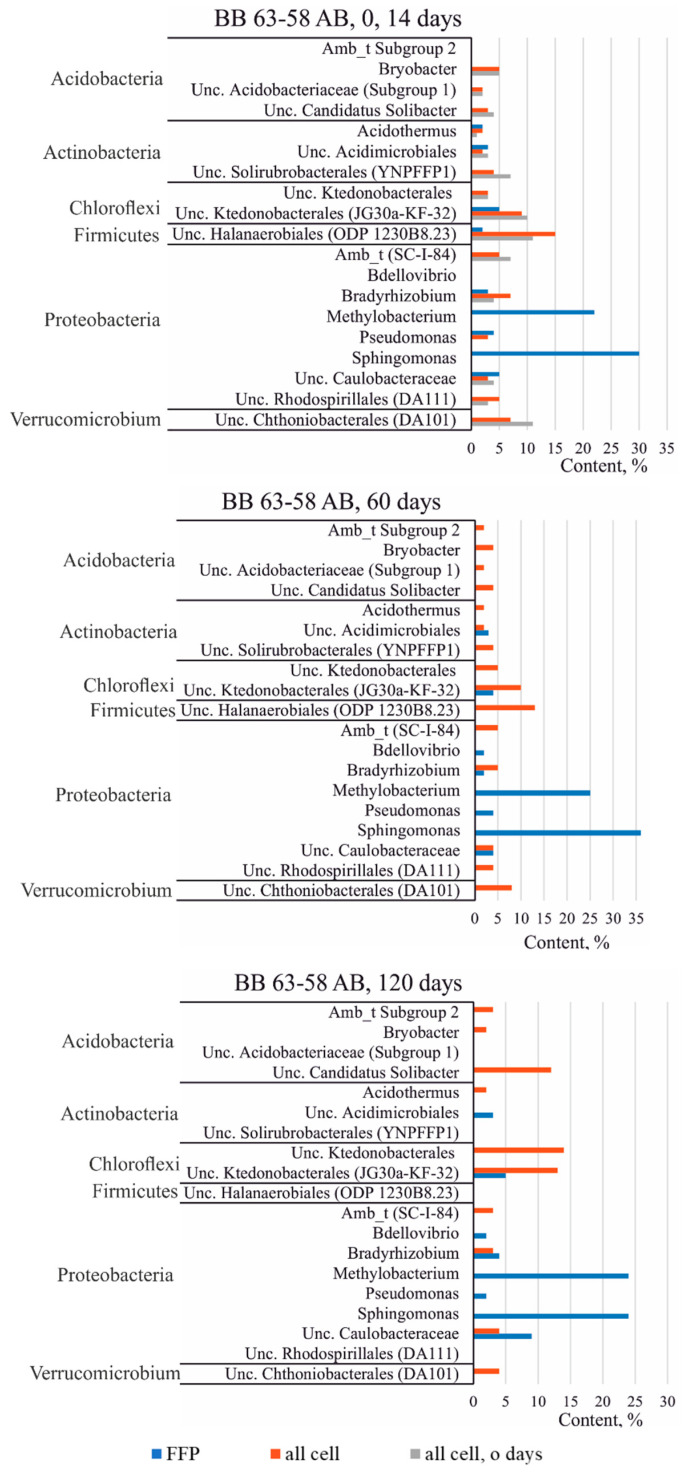
Distribution of dominant groups during succession in the horizon AB, BB 63–58 profile, 20 °C (phyla with content greater than 2% are presented).

**Figure 8 microorganisms-09-01728-f008:**
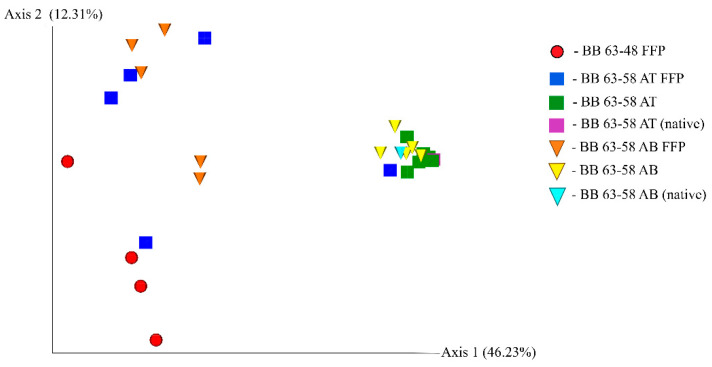
Two-dimensional diagram of the phylogenetic similarity of bacterial communities of the horizons of the studied Antarctic soils according to the results of unweighted UniFrac.

**Table 1 microorganisms-09-01728-t001:** Description of the samples.

Sampling Location	Soil Profile, №	Horizon, Depth in [cm]	C	N	W
			[%]		
The central part of the Hills, flat valley, fragmentary surface sodium bicarbonate crusts, Typic Haploturbel without an organogenic horizon	BB 63–48	10–15	0.13 ± 0.00	0.03 ± 0.00	4.17 ± 0.15
Southern part of the Hills, wind shelter, Typic Haplorthel with an organogenic horizon	BB 63–58	AT, 0–2	6.62 ± 0.01	0.32 ± 0.01	10.02 ± 0.11
		AB, 10–15	3.0 ± 0.01	0.40 ± 0.01	6.49 ± 0.20

C—Total carbon; N—Total nitrogen; W—Humidity of soil.

**Table 2 microorganisms-09-01728-t002:** The total content of carbon, nitrogen in the studied samples.

Sample	C [%]			N [%]		
	Native	At the End of Succession (120 Days)		Native	At the End of Succession (120 Days)	
		5 °C	20 °C		5 °C	20 °C
BB 63–48	0.13	0.12	0.18	0.03	0.08	0.06
BB 63–58 (AT)	6.63	4.97	6.38	0.66	0.50	0.71
BB 63–58 (AB)	3.03	4.12	2.90	0.39	0.44	0.39

**Table 3 microorganisms-09-01728-t003:** The Simpson Index calculated based on the number of sequences along the course of succession in the studied samples of Antarctic soils.

Sample			Succession Days						
			0	14		60		120	
				5 °C	20 °C	5 °C	20 °C	5 °C	20 °C
BB 63–48		Soil	-	-	-	-	-	-	-
		FFB	-	-	-	0.913	0.909	0.891	0.912
BB 63–58	AT	Soil	0.966	0.962	0.973	0.972	0.961	0.969	0.970
		FFB	-	0.949	0.972	0.919	0.939	0.923	-
	AB	Soil	0.972	0.970	0.969	0.972	0.969	-	0.954
		FFB	-	0.957	0.945	0.932	0.918	-	0.947

- not enough DNA was isolated for next-generation sequencing.

**Table 4 microorganisms-09-01728-t004:** Shannon Index calculated based on the number of sequences along the course of succession in the studied samples of Antarctic soils.

Sample			Succession Days						
			0	14		60		120	
				5 °C	20 °C	5 °C	20 °C	5 °C	20 °C
BB 63–48		Soil	-	-	-	-	-	-	-
		FFB	-	-	-	5.4	5.2	4.6	4.8
BB 63–58	AT	Soil	6.5	6.4	6.6	6.5	6.2	6.7	6.7
		FFB	-	5.7	6.8	5.1	5.3	5.1	-
	AB	Soil	6.4	6.4	6.3	6.4	6.3	-	6.0
		FFB	-	6.3	5.8	5.5	5.4	-	5.6

- not enough DNA was isolated for next-generation sequencing.

**Table 5 microorganisms-09-01728-t005:** Alphaproteobacteria/Acidobacteria ratio.

Temperature	5 °C				20 °C			
Days of Succession	0	14	60	120	0	14	60	120
BB 63–58 (AT)								
all bacteria	0.71	0.82	0.70	0.60	0.71	0.84	0.73	0.62
FFB fraction	-	12.5	85.0	130.0	-	2.25	28.5	-
BB 63–58 (AB)								
all bacteria	0.83	0.76	0.78	-	0.83	1.43	0.89	0.71
FFB fraction	-	12.6	33.5	-	-	16.5	18.0	13.4

«-»—not enough DNA was isolated for next-generation sequencing.

## Data Availability

The data presented in this study are available on request from the corresponding authors.

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
