# Peer review of "The Abundance and Taxonomic Diversity of Filterable Forms of Bacteria during Succession in the Soils of Antarctica (Bunger Hills)"

_microorganisms, 2021, doi:10.3390/microorganisms9081728_

Round 1

Reviewer 1 Report

The paper is very interesting. I suggest the following modifications. Introduction: I suggest better enphasize the importance to study the filterable forms of bacteria (FFB). Methods:in my knowledge, a triplicate of each sample must be performed to have a robust statistical analysis of results. Results: fig. 4. The standard deviation from which data come from? Discussion: why the authors do not compare their results with those reported for a bacterial consortium associated with an Antarctic ciliate (Pucciarelli, S., Devaraj, R.R., Mancini, A. et al. Microbial Consortium Associated with the Antarctic Marine Ciliate Euplotes focardii: An Investigation from Genomic Sequences. Microb Ecol 70, 484–497 (2015). https://doi.org/10.1007/s00248-015-0568-9)? This may help to discuss the role of the bacterial phyla and cold adaptation. Furthermore, the discussion is too long, with several repetition. I suggest reducing this part

Author Response

The paper is very interesting. I suggest the following modifications. Introduction: I suggest better enphasize the importance to study the filterable forms of bacteria (FFB).

- Thank you for your interest in our work and useful comments. We tried to additionally emphasize the importance of studying the filtered forms of bacteria – Lines 86-93.

Methods:in my knowledge, a triplicate of each sample must be performed to have a robust statistical analysis of results.

With a direct calculation of the total number of cells and the number of FFB, this was done (lines 250-251). Unfortunately, during NGS analysis of DNA, it was impossible to analyze three probes at each checkpoint due to the small number of initial soil samples and the low content of cells in them. However, the fact that the results obtained on three samples did not differ very much indicates the reliability of the results obtained.

Results: fig. 4. The standard deviation from which data come from?

- The total number of bacteria and the number of the FFB were determined using fluorescence microscopy using the dye acridine orange (lines 245-258). To clarify this point we made the separate section in Materials and methods.

Discussion: why the authors do not compare their results with those reported for a bacterial consortium associated with an Antarctic ciliate (Pucciarelli, S., Devaraj, R.R., Mancini, A. et al. Microbial Consortium Associated with the Antarctic Marine Ciliate Euplotes focardii: An Investigation from Genomic Sequences. Microb Ecol 70, 484–497 (2015). https://doi.org/10.1007/s00248-015-0568-9)? This may help to discuss the role of the bacterial phyla and cold adaptation.

Thank you for your comment. This is really a very interesting work, but it seems to us that its subject matter is not related to the tasks set in our study and the conclusions that follow from it.

 Furthermore, the discussion is too long, with several repetition. I suggest reducing this part

Thanks. It was done (lines 734 – 742).

Reviewer 2 Report

L10-11: who stated? on what basis? terribly general sentence. Perhaps it would be better to write something like: "previous studies have shown..."

L11-12: the sentence should be reworded. rather in the style that previous work/studies aroused interest, or on their basis new research questions were raised, etc.

Abstract and the rest of the paper: once it is written in impersonal form that something was done (e.g. L 16-18) and once in personal form that we did something (e.g. L13-14; L43-44). Please standardize this throughout the manuscript. I suggest in scientific papers the impersonal form.

 L29-30: references?

L48-51: references?

L60-64: this fragment should be extracted and some additional information and clear reference to your own work should be added, and L65-75 should be left separately.

L82-83: references

Figure 1b: source of photographs

 L94-101: this continues the description of the research material, including Table 1. I suggest making Chapter 2. "material and methods"; 2.1. "soil samples;" and so on. "Scheme of the experiment to initiate succession" - title inadequate for content. I suggest changing to "physico-chemical analyses of samples" or something like that.

Table 1: C, N – explain abbreviations in detail - what carbon and what nitrogen? total? organic? specifically. Units should be in square brackets i.e. [%] and [cm] in the horizon column.

L139: what DNA analysis? Here is only electrophoresis. For what purpose performed?

L152: please provide the primer sequences.

L147-172: here is a mixed order, the description of statistical analyses and bionformat processing should be the last subsection in the methodology. Besides, section 2.2.4 and 2.2.6 look almost identical, in 2.2.4 it should be made clear that it is about total DNA from soil samples - both in the title of the subsection and in the text; it is best to make a point "isolation of total soil DNA and NGS of the 16S rRNA gene" and "isolation and analysis of DNA from bacetrial isolates". Because at the moment there is confusion.

L173: please provide the primer sequences.

L181: and have sequences been deposited in the database?

L186: the markers of horizon (AT, AB etc.) should be entered in table 1

L207: clarify EPS

L214: add sp. after Actionobacteria

L222-224: this is more for discussion

Figure 4: axes poorly legible; where is the statistical analysis?

Table 2: units in []; where statistical analysis?

L284-286; 296-312 And further: names Kingdom, phylum, class, order, and suborder begin with a capital letter but are not italicized.

Figure 5: poor resolution; no captions on Y axis.

L321-328; 333-339; 363-366; 376-378; 384-387, 450-456, 467-469: these passages fit the discussion, not the presentation of results

Figure 6: illegible

Table 3, 4, 5: please format it correctly, because it is illegible in the present grease. Statistical analysis? If too little DNA was obtained for NGS then this should be written already in the methodology.

Figure 7: why are the axes signed "axis"? what do they represent? where is the unit and scale?

L473: CFU - explain

Why is there no statistical analysis anywhere?

Author Response

Comments and Suggestions for Authors

Answer: Thank you for your interest in our work and useful comments.

1 )

L10-11: who stated? on what basis? terribly general sentence. Perhaps it would be better to write something like: "previous studies have shown..."

L11-12: the sentence should be reworded. rather in the style that previous work/studies aroused interest, or on their basis new research questions were raised, etc.

Abstract and the rest of the paper: once it is written in impersonal form that something was done (e.g. L 16-18) and once in personal form that we did something (e.g. L13-14; L43-44). Please standardize this throughout the manuscript. I suggest in scientific papers the impersonal form.

Answer: Thank you for useful comments, we tried to correct it (L 52-72, L 64-65). The line numbers here and below are for the "Show all changes" mode.

2)

 1)L29-30: references?

2)L48-51: references?

3)L82-83: references ????

Answer: 1) We added the references (L 77-78). 2) This is our assumption based on the analysis of the literature (L 57-60). 3) The samples used in this work were taken by Andrey Dolgih, coauthor of the present work, and have not been described anywhere before. We have tried to clarify this point (L 111).  Thank you.

3)

L60-64: this fragment should be extracted and some additional information and clear reference to your own work should be added, and L65-75 should be left separately.

Answer:

Thank you for your suggestion. We added references to our articles (L 78).  But we decided not to change this fragment, because on the basis of the previously published data described in it, we conceived and planned the present work.

4)

Figure 1b: source of photographs

 L94-101: this continues the description of the research material, including Table 1. I suggest making Chapter 2. "material and methods"; 2.1. "soil samples;" and so on. "Scheme of the experiment to initiate succession" - title inadequate for content. I suggest changing to "physico-chemical analyses of samples" or something like that.

Table 1: C, N – explain abbreviations in detail - what carbon and what nitrogen? total? organic? specifically. Units should be in square brackets i.e. [%] and [cm] in the horizon column.

L152: please provide the primer sequences.

L147-172: here is a mixed order, the description of statistical analyses and bionformat processing should be the last subsection in the methodology. Besides, section 2.2.4 and 2.2.6 look almost identical, in 2.2.4 it should be made clear that it is about total DNA from soil samples - both in the title of the subsection and in the text; it is best to make a point "isolation of total soil DNA and NGS of the 16S rRNA gene" and "isolation and analysis of DNA from bacetrial isolates". Because at the moment there is confusion.

L173: please provide the primer sequences.

L186: the markers of horizon (AT, AB etc.) should be entered in table 1

L207: clarify EPS

L214: add sp. after Actionobacteria

L222-224: this is more for discussion

L473: CFU - explain

Answer:

Thanks for the comments. We have made all the corrections (L 121-126, 138, 139, 212-213, 235-236, 207, 233, 246-254, 284, 292, 590-592, 567). The markers of horizon (AT, AB) were entered in Table 1 (the “Horizon, depth in [cm]” column).

5)

L139: what DNA analysis? Here is only electrophoresis. For what purpose performed?  ????

Answer:

Electrophoresis was used to assess the quality and size of the isolated DNA and PCR products. We decided to remove this section from methods, as it is a helper method (L 199-206).

6)

L181: and have sequences been deposited in the database?

Answer:

The sequences obtained using the NGS method were entered into the database. This is mentioned in the article (L 256-259).

7)

Figure 4: axes poorly legible; where is the statistical analysis?

Table 2: units in []; where statistical analysis?

Figure 5: poor resolution; no captions on Y axis

Figure 6: illegible

Table 3, 4, 5: please format it correctly, because it is illegible in the present grease. Statistical analysis? If too little DNA was obtained for NGS then this should be written already in the methodology.

Why is there no statistical analysis anywhere?

Answer:

Thank you. We tried to improve the quality of all figures.

Statistical analysis was performed for all data for which it was possible to obtain results in more than one repetition (Box-and-whiskers in Figure 4).

We have added information about the amount of total DNA isolated from the samples to the manuscript (L 359-362).

Sequencing was carried out in one repetition. During NGS analysis of DNA, it was impossible to analyze more than one probe at each checkpoint of succession due to the small number of original soil samples and their low cell content. However, the fact that the results obtained with the three samples did not differ greatly indicates the reliability of the results obtained.

8)

L284-286; 296-312 And further: names Kingdom, phylum, class, order, and suborder begin with a capital letter but are not italicized.

Answer:

The names of Kingdom, phylum, class, order, and suborder should be written in Latin, which should be italicized in English text according to the naming and spelling rules for bacteria taxa governed by the International Code of Nomenclature of Prokaryotes (https://doi.org/10.1099/ijsem.0.000778).

9)

L321-328; 333-339; 363-366; 376-378; 384-387, 450-456, 467-469: these passages fit the discussion, not the presentation of results

Answer:

Thank you. We agree with you that these lines are more suitable for the "Discussion" section, but we decided to leave them in the "Results" section, since they briefly explain the results obtained and they explain why we decided to pay attention to one or another group of bacteria. Carrying over these short explanations of the results will lead to a large increase in the article and "Discussion" section and will add repetitions of information. In the Discussion section, we have tried to summarize the information and to give more general assumptions and conclusions.

10) Figure 7: why are the axes signed "axis"? what do they represent? where is the unit and scale?

Answer:

The figure shows the result of an unweight Unifrac analysis in the form of a two-dimensional principal component analysis. This is not a parametric method of analysis, is a distance metric used for comparing biological communities. It incorporates information on the relative relatedness of community members by incorporating phylogenetic distances between observed organisms in the computation. Percentages indicate how many percent of differences in the taxonomic structure of communities are explained by each of the components.

We used one of the accepted and used forms of presenting the results of such an analysis. See for example: Yergeau E, Bokhorst S, Kang S, Zhou J, Greer CW, Aerts R, Kowalchuk GA. Shifts in soil microorganisms in response to warming are consistent across a range of Antarctic environments. ISME J. 2012 Mar;6(3):692-702. doi: 10.1038/ismej.2011.124. Epub 2011 Sep 22. PMID: 21938020; PMCID: PMC3282189.

Round 2

Reviewer 2 Report

Thank you for referring to my comments. I accept the amendments made.